# UniFormer: Unified Transformer for Efficient Spatiotemporal Representation Learning

**Kunchang Li**[123],[*] **Yali Wang**[1],[*] **Peng Gao**[3], **Guanglu Song**[4]
**Yu Liu**[4], **Hongsheng Li** [5], **Yu Qiao**[13][†]
[1]ShenZhen Key Lab of Computer Vision and Pattern Recognition, SIAT-SenseTime Joint Lab,
Shenzhen Institute of Advanced Technology, Chinese Academy of Sciences
[2]University of Chinese Academy of Sciences, [3]Shanghai AI Laboratory, Shanghai, China
[4]SenseTime Research, [5]The Chinese University of Hong Kong
{kc.li,yl.wang}@siat.ac.cn, {gaopeng,qiaoyu}@pjlab.org.cn
songguanglu@sensetime.com, liuyuisanai@gmail.com
hsli@ee.cuhk.edu.hk

## Abstract

It is a challenging task to learn rich and multi-scale spatiotemporal semantics from high-dimensional videos, due to large local redundancy and complex global dependency between video frames. The recent advances in this research have been mainly driven by 3D convolutional neural networks and vision transformers. Although 3D convolution can efficiently aggregate local context to suppress local redundancy from a small 3D neighborhood, it lacks the capability to capture global dependency because of the limited receptive field. Alternatively, vision transformers can effectively capture long-range dependency by self-attention mechanism, while having the limitation on reducing local redundancy with blind similarity comparison among all the tokens in each layer. Based on these observations, we propose a novel Unified transFormer (UniFormer) which seamlessly integrates merits of 3D convolution and spatiotemporal self-attention in a concise transformer format, and achieves a preferable balance between computation and accuracy. Different from traditional transformers, our relation aggregator can tackle both spatiotemporal redundancy and dependency, by learning local and global token affinity respectively in shallow and deep layers. We conduct extensive experiments on the popular video benchmarks, e.g., Kinetics-400, Kinetics-600, and Something-Something V1&V2. With only ImageNet-1K pretraining, our UniFormer achieves 82.9%/84.8% top-1 accuracy on Kinetics-400/Kinetics-600, while requiring $10\times$ fewer GFLOPs than other state-of-the-art methods. For Something-Something V1 and V2, our UniFormer achieves new state-of-the-art performances of 60.9% and 71.2% top-1 accuracy respectively. Code is available at https://github.com/Sense-X/UniFormer.

## 1 Introduction

Learning spatiotemporal representations is a fundamental task for video understanding. Basically, there are two distinct challenges. On the one hand, videos contain large spatiotemporal redundancy, where target motions across local neighboring frames are subtle. On the other hand, videos contain complex spatiotemporal dependency, since target relations across long-range frames are dynamic.

The advances in video classification have mostly driven by 3D convolutional neural networks (Tran et al., 2015; Carreira & Zisserman, 2017b; Feichtenhofer et al., 2019) and spatiotemporal transformers (Bertasius et al., 2021; Arnab et al., 2021). Unfortunately, each of these two frameworks focuses on one of the aforementioned challenges. 3D convolution can capture detailed and local spatiotemporal features, by processing each pixel with context from a small 3D neighborhood (e.g., $3\times3\times3$).

---

[*]Equally-contributed first authors ({kc.li, yl.wang}@siat.ac.cn)
[†]Corresponding author (qiaoyu@pjlab.org.cn)

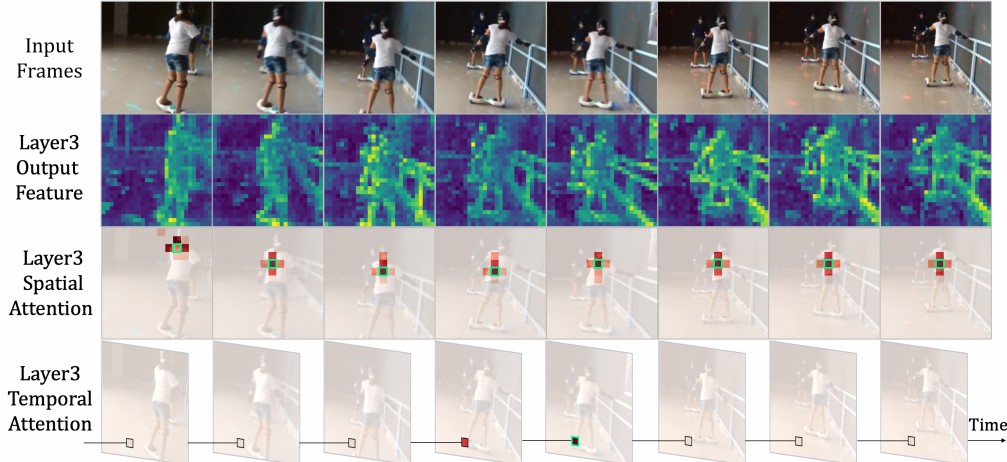

Figure 1: **Some visualizations of TimeSformer.** We respectively show the feature, spatial and temporal attention from the 3rd layer of TimeSformer (Bertasius et al., 2021). We find that, such transformer learns local representations with redundant global attention. For an anchor token (green box), spatial/temporal attention compares it with all the contextual tokens for aggregation, while only its neighboring tokens (boxes filled with red color) actually work. Hence, it wastes large computation to encode only very local spatiotemporal representations.

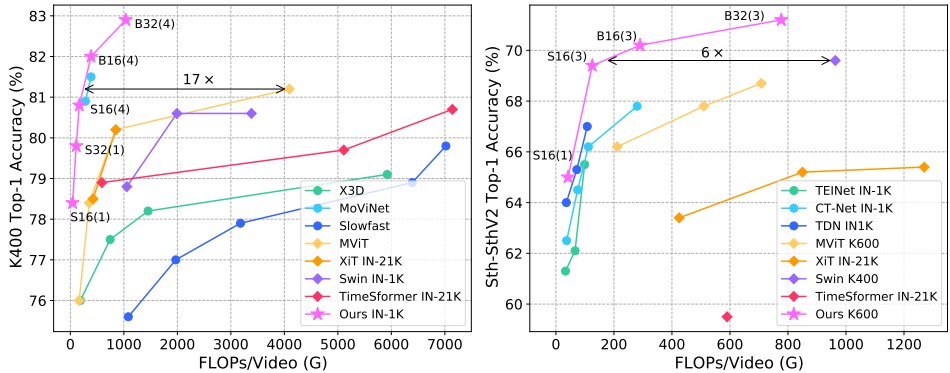

Figure 2: **Accuracy vs. per-video GFLOPs on Kinetics-400 and Something-Something V2.** B-32(4) means we test UniFormer-$B_{32f}$ with 4 clips and S-16(3) means we test UniFormer-$S_{16f}$ with 3 crops (more testing details can be found in Section 4.3). Our UniFormer achieves the best balance between accuracy and computation on both datasets.

Hence, it can reduce spatiotemporal redundancy across adjacent frames. However, due to the limited receptive field, 3D convolution suffers from difficulty in learning long-range dependency (Wang et al., 2018; Li et al., 2020a). Alternatively, vision transformers are good at capturing global dependency, with the help of self-attention among visual tokens (Dosovitskiy et al., 2021). Recently, this design has been introduced in video classification via spatiotemporal attention mechanism (Bertasius et al., 2021). However, we observe that, video transformers are often inefficient to encode local spatiotemporal features in the shallow layers. We take the well-known and typical TimeSformer (Bertasius et al., 2021) for illustration. As shown in Figure 1, TimeSformer indeed learns detailed video representations in the early layers, but with very redundant spatial and temporal attention. Specifically, spatial attention mainly focuses on the neighbor tokens (mostly in 3×3 local regions), while learning nothing from the rest tokens in the same frame. Similarly, temporal attention mostly only aggregates tokens in the adjacent frames, while ignoring the rest in the distant frames. More importantly, such local representations are learned from global token-to-token similarity comparison in all layers, requiring large computation cost. This fact clearly deteriorates computation-accuracy balance of such video transformer (Figure 2).

To tackle these difficulties, we propose to effectively unify 3D convolution and spatiotemporal self-attention in a concise transformer format, thus we name the network Unified transFormer (Uni-Former), which can achieve a preferable balance between efficiency and effectiveness. More specifically, our UniFormer consists of three core modules, i.e., Dynamic Position Embedding (DPE),

Multi-Head Relation Aggregator (MHRA), and Feed-Forward Network (FFN). The key difference between our UniFormer and traditional video transformers is the distinct design of our relation aggregator. First, instead of utilizing a self-attention mechanism in all layers, our proposed relation aggregator tackles video redundancy and dependency respectively. In the shallow layers, our aggregator learns local relation with a small learnable parameter matrix, which can largely reduce computation burden by aggregating context from adjacent tokens in a small 3D neighborhood. In the deep layers, our aggregator learns global relation with similarity comparison, which can flexibly build long-range token dependencies from distant frames in the video. Second, different from spatial and temporal attention separation in the traditional transformers (Bertasius et al., 2021; Arnab et al., 2021), our relation aggregator jointly encodes spatiotemporal context in all the layers, which can further boost video representations in a joint learning manner. Finally, we build up our model by progressively integrating UniFormer blocks in a hierarchical manner. In this case, we enlarge the cooperative power of local and global UniFormer blocks for efficient spatiotemporal representation learning in videos. We conduct extensive experiments on the popular video benchmarks, e.g., Kineticss-400 (Carreira & Zisserman, 2017a), Kinetics-600 (Carreira et al., 2018) and Something-Something V1&V2 (Goyal et al., 2017b). With only ImageNet-1K pretraining, our UniFormer achieves 82.9%/84.8% top-1 accuracy on Kinetics-400/Kinetics-600, while requiring $10\times$ fewer GFLOPs than other comparable methods (e.g., $16.7\times$ fewer GFLOPs than ViViT (Arnab et al., 2021) with JFT-300M pre-training). For Something-Something V1 and V2, our UniFormer achieves 60.9% and 71.2% top-1 accuracy respectively, which are new state-of-the-art performances.

## 2    RELATED WORK

**Convolution-based Video Networks.** 3D Convolution Neural Networks (CNNs) have been dominant in video understanding (Tran et al., 2015; Feichtenhofer et al., 2019). However, they suffer from the difficult optimization problem and large computation cost. To resolve this issue, I3D (Carreira & Zisserman, 2017b) inflates the pre-trained 2D convolution kernels for better optimization. Other prior works (Tran et al., 2018; Qiu et al., 2017; Tran et al., 2019; Feichtenhofer, 2020; Wang et al., 2020a) try to factorize 3D convolution kernel in different dimensions to reduce complexity. Recent methods propose well-designed modules to enhance the temporal modeling ability for 2D CNNs (Wang et al., 2016; Lin et al., 2019; Luo & Yuille, 2019; Jiang et al., 2019; Liu et al., 2020a; Li et al., 2020b; Kwon et al., 2020; Li et al., 2020a; 2021a; Wang et al., 2020b). However, 3D convolution struggles to capture long-range dependency, due to the limited receptive field.

**Transformer-based Video Networks.** Vision Transformers (Dosovitskiy et al., 2021; Touvron et al., 2021a;b; Liu et al., 2021a) have been popular for vision tasks and outperform many CNNs. Based on ViT, several prior works (Bertasius et al., 2021; Neimark et al., 2021; Sharir et al., 2021; Li et al., 2021b; Arnab et al., 2021; Bulat et al., 2021; Patrick et al., 2021; Zha et al., 2021) propose different variants for spatiotemporal learning, verifying the outstanding ability of the transformer to capture long-term dependencies. To reduce high dot-product computation, MViT (Fan et al., 2021) introduces the hierarchical structure and pooling self-attention, while Video Swin (Liu et al., 2021b) advocates an inductive bias of locality for video. Nevertheless, the self-attention mechanism is inefficient to encode low-level features, hindering their high potential. To tackle this challenge, different from Video Swin that applies self-attention in a local 3D window, we adopt 3D convolution in a concise transformer format to encode local features. Besides, we follow their hierarchical designs and propose our UniFormer, achieving powerful performance for video understanding.

## 3    METHOD

In this section, we describe our UniFormer in detail. First, we introduce the overall architecture of the UniFormer block. Then, we explain the vital designs of our UniFormer for spatiotemporal modeling, i.e., multi-head relation aggregator and dynamic position embedding. Finally, we hierarchically stack UniFormer blocks to build up our video network.

### 3.1    OVERVIEW OF UNIFORMER BLOCK

To overcome problems of spatiotemporal redundancy and dependency, we propose a novel and concise Unified transFormer (UniFormer) shown in Figure 3. We utilize a basic transformer format

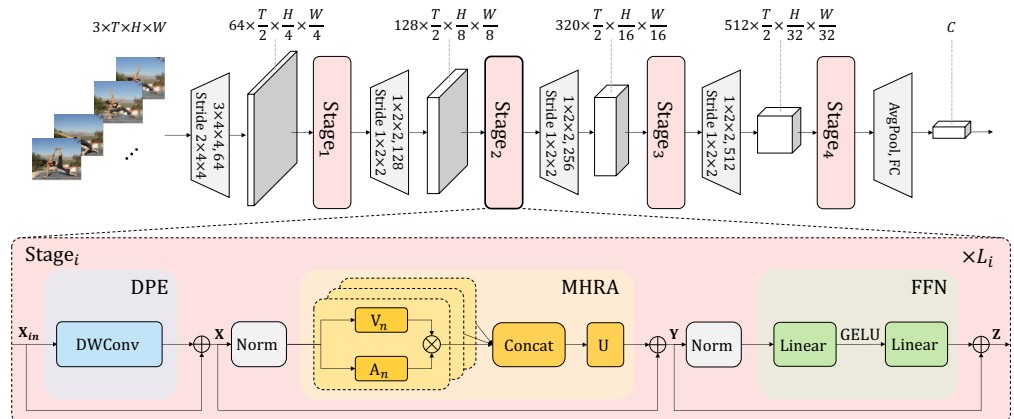

Figure 3: Overall architecture of our Unified transFormer (UniFormer). A UniFormer block consists of three key modules, i.e., Dynamic Position Embedding (DPE), Multi-Head Relation Aggregrator (MHRA), and Feed Forward Network (FFN). Detailed explanations can be found in Section 3.

(Vaswani et al., 2017) but specially design it for efficient and effective spatiotemporal representation learning. Specifically, our UniFormer block consists of three key modules: Dynamic Position Embedding (DPE), Multi-Head Relation Aggregator (MHRA), and Feed-Forward Network (FFN):

$$\mathbf{X} = \mathrm{DPE}\left(\mathbf{X}_{in}\right) + \mathbf{X}_{in}, \tag{1}$$

$$\mathbf{Y} = \mathrm{MHRA}\left(\mathrm{Norm}\left(\mathbf{X}\right)\right) + \mathbf{X}, \tag{2}$$

$$\mathbf{Z} = \mathrm{FFN}\left(\mathrm{Norm}\left(\mathbf{Y}\right)\right) + \mathbf{Y}. \tag{3}$$

Considering the input token tensor (frame volumes) $\mathbf{X}_{in} \in \mathbb{R}^{C \times T \times H \times W}$, we first introduce DPE to dynamically integrate 3D position information into all the tokens (Eq. 1), which effectively makes use of spatiotemporal order of the tokens for video modeling. Then, we leverage MHRA to aggregate each token with its contextual tokens (Eq. 2). Different from the regular Multi-Head Self-Attention (MHSA), our MHRA smartly tackles local video redundancy and global video dependency, by flexible designs of token affinity learning in the shallow and deep layers. Finally, we add FFN with two linear layers for pointwise enhancement of each token (Eq. 3).

## 3.2 MULTI-HEAD RELATION AGGREGATOR

As discussed above, we should solve large local redundancy and complex global dependency, for efficient and effective spatiotemporal representation learning. Unfortunately, the popular 3D CNNs and spatiotemporal transformers only focus on one of these two challenges. For this reason, we design an alternative Relation Aggregator (RA), which can flexibly unify 3D convolution and spatiotemporal self-attention in a concise transformer format, solving video redundancy and dependency in the shallow layers and deep layers respectively. Specifically, our MHRA conducts token relation learning via multi-head fusion:

$$\mathrm{R}_n(\mathbf{X}) = \mathrm{A}_n \mathrm{V}_n(\mathbf{X}), \tag{4}$$

$$\mathrm{MHRA}(\mathbf{X}) = \mathrm{Concat}(\mathrm{R}_1(\mathbf{X}); \mathrm{R}_2(\mathbf{X}); \cdots ; \mathrm{R}_N(\mathbf{X}))\mathbf{U}. \tag{5}$$

Given the input tensor $\mathbf{X} \in \mathbb{R}^{C \times T \times H \times W}$, we first reshape it to a sequence of tokens $\mathbf{X} \in \mathbb{R}^{L \times C}$, $L = T \times H \times W$. $\mathrm{R}_n(\cdot)$ is the relation aggregator (RA) in the $n$-th head, and $\mathbf{U} \in \mathbb{R}^{C \times C}$ is a learnable parameter matrix to integrate $N$ heads. Moreover, each RA consists of token context encoding and token affinity learning. Via a linear transformation, one can transform the original token into context $\mathrm{V}_n(\mathbf{X}) \in \mathbb{R}^{L \times \frac{C}{N}}$. Subsequently, the relation aggregator can summarize context with the guidance of the token affinity $\mathrm{A}_n \in \mathbb{R}^{L \times L}$. The key in our RA is how to learn $\mathrm{A}_n$ in videos.

**Local MHRA.** In the shallow layers, we aim at learning detailed video representation from the local spatiotemporal context in small 3D neighborhoods. This coincidentally shares a similar insight with the design of a 3D convolution filter. As a result, we design the token affinity as a learnable parameter matrix operated in the local 3D neighborhood, i.e., given one anchor token $\mathbf{X}_i$, RA learns local spatiotemporal affinity between this token and other tokens in the small tube $\Omega_i^{t \times h \times w}$:

$$\mathrm{A}_n^{local}(\mathbf{X}_i, \mathbf{X}_j) = a_n^{i-j}, \ \ where \ j \in \Omega_i^{t \times h \times w}, \tag{6}$$

| Method | Basic Operation | Tackle Local Redundancy | Capture Global Dependency | Efficiency GFLOPs | Top-1 |
|---|---|---|---|---|---|
| X3D (Feichtenhofer, 2020) | PWConv-DWConv-PWConv | ✔ | ✗ | 5823 | 80.4 |
| TimeSformer (Bertasius et al., 2021) | Divided MHSA | ✗ | ✔ | 7140 | 80.7 |
| Our UniFormer | Joint MHRA | ✔ | ✔ | **168** | **80.8** |

Table 1: **Comparison to different methods.** 'Local Redundancy' means the redundant computation for capturing local features. 'Global Dependency' means the long-range dependency among frames.

where $a_n \in \mathbb{R}^{t \times h \times w}$ and $\mathbf{X}_j$ refers to any neighbor token in $\Omega_i^{t \times h \times w}$. $(i - j)$ means the relative token index that determines the aggregating weight (Appendix A shows more details). In the shallow layers, video contents between adjacent tokens vary subtly, it is significant to encode detailed features with local operator to reduce redundancy. Hence, the token affinity is designed as a local learnable parameter matrix, whose values only depend on the relative 3D position between tokens.

**Comparison to 3D Convolution Block.** Interestingly, we find that our local MHRA can be interpreted as a spatiotemporal extension of MobileNet block (Sandler et al., 2018; Tran et al., 2019; Feichtenhofer, 2020). Specifically, the linear transformation $\mathrm{V}(\cdot)$ can be instantiated as pointwise convolution (PWConv). Furthermore, the local token affinity $\mathrm{A}_n^{local}$ is a spatiotemporal matrix that operated on each output channel (or head) $\mathrm{V}_n(\mathbf{X})$, thus the relation aggregator $\mathrm{R}_n(\mathbf{X}) = \mathrm{A}_n^{local}\mathrm{V}_n(\mathbf{X})$ can be explained as a depthwise convolution (DWConv). Finally, all heads are concatenated and fused by a linear matrix $\mathbf{U}$, which can also be instantiated as pointwise convolution (PWConv). As a result, this local MHRA can be reformulated with a manner of PWConv-DWConv-PWConv in the MobileNet block. In our experiments, we flexibly instantiate our local MHRA as such channel-separated spatiotemporal convolution, so that our UniFormer can inherit computation efficiency for light-weight video classification. Different from the MobileNet block, our UniFormer block is designed as a generic transformer format, thus an extra FFN is inserted after MHRA, which can further mix token context at each spatiotemporal position to boost classification accuracy.

**Global MHRA.** In the deep layers, we focus on capturing long-term token dependency in the global video clip. This naturally shares a similar insight with the design of self-attention. Hence, we design the token affinity via comparing content similarity among all the tokens in the global view:

$$\mathrm{A}_n^{global}(\mathbf{X}_i, \mathbf{X}_j) = \frac{e^{Q_n(\mathbf{X}_i)^T K_n(\mathbf{X}_j)}}{\sum_{j' \in \Omega_{T \times H \times W}} e^{Q_n(\mathbf{X}_i)^T K_n(\mathbf{X}_{j'})}}, \tag{7}$$

where $\mathbf{X}_j$ can be any token in the global 3D tube with size of $T \times H \times W$, while $Q_n(\cdot)$ and $K_n(\cdot)$ are two different linear transformations. Most video transformers apply self-attention in all stages, introducing a large amount of calculation. To reduce the dot-product computation, the prior works tend to divide spatial and temporal attention (Bertasius et al., 2021; Arnab et al., 2021), but it deteriorates the spatiotemporal relation among tokens. In contrast, our MHRA performs local relation aggregation in the early layers, which largely saves the computation of token comparison. Hence, instead of factorizing spatiotemporal attention, we jointly encode spatiotemporal relation in our MHRA for all the stages, in order to achieve a preferable computation-accuracy balance.

**Comparison to Transformer Block**. In the deep layers, our UniFormer block is equipped with a global MHRA $\mathrm{A}_n^{global}$ (Eq. 7). It can be instantiated as a spatiotemporal self attention, where $Q_n(\cdot)$, $\mathrm{K}_n(\cdot)$ and $\mathrm{V}_n(\cdot)$ become Query, Key and Value in the transformer (Dosovitskiy et al., 2021). Hence, it can effectively learn long-term dependency. Instead of spatial and temporal factorization in the previous video transformers (Bertasius et al., 2021; Arnab et al., 2021), our global MHRA is based on joint spatiotemporal learning to generate more discriminative video representation. Moreover, we adopt dynamic position embedding (DPE, see Section 3.3) to overcome permutation-invariance, which can maintain translation-invariance and is friendly to different input clip lengths.

### 3.3 DYNAMIC POSITION EMBEDDING

Since videos are both spatial and temporal variant, it is necessary to encode spatiotemporal position information for token representations. The previous methods mainly adapt the absolute or relative position embedding of image tasks to tackle this problem (Bertasius et al., 2021; Arnab et al., 2021). However, when testing with longer input clips, the absolute one should be interpolated to target input size with fine-tuning. Besides, the relative version modifies the self-attention and performs worse due to lack of absolute position information (Islam et al., 2020). To overcome the above problems,

| Method | Pretrain | #Frame | GFLOPs | K400 Top-1 | K400 Top-5 | K600 Top-1 | K600 Top-5 |
|---|---|---|---|---|---|---|---|
| LGD(Qiu et al., 2019) | IN-1K | 128×N/A | N/A | 79.4 | 94.4 | 81.5 | 95.6 |
| SlowFast+NL(Feichtenhofer et al., 2019) | - | 16×3×10 | 7020 | 79.8 | 93.9 | 81.8 | 95.1 |
| ip-CSN(Tran et al., 2019) | Sports1M | 32×3×10 | 3270 | 79.2 | 93.8 | - | - |
| CorrNet(Wang et al., 2020a) | Sports1M | 32×3×10 | 6720 | 81.0 | - | - | - |
| X3D-M(Feichtenhofer, 2020) | - | 16×3×10 | 186 | 76.0 | 92.3 | 78.8 | 94.5 |
| X3D-XL(Feichtenhofer, 2020) | - | 16×3×10 | 1452 | 79.1 | 93.9 | 81.9 | 95.5 |
| MoViNet-A5(Kondratyuk et al., 2021) | - | 120×1×1 | 281 | 80.9 | 94.9 | 82.7 | 95.7 |
| MoViNet-A6(Kondratyuk et al., 2021) | - | 120×1×1 | 386 | 81.5 | 95.3 | 83.5 | 96.2 |
| ViT-B-VTN (Neimark et al., 2021) | IN-21K | 250×1×1 | 3992 | 78.6 | 93.7 | - | - |
| TimeSformer-L(Bertasius et al., 2021) | IN-21K | 96×3×1 | 7140 | 80.7 | 94.7 | 82.2 | 95.5 |
| STAM (Sharir et al., 2021) | IN-21K | 64×1×1 | 1040 | 79.2 | - | - | - |
| X-ViT(Bulat et al., 2021) | IN-21K | 16×3×1 | 850 | 80.2 | 94.7 | 84.5 | 96.3 |
| Mformer-HR(Patrick et al., 2021) | IN-21K | 16×3×10 | 28764 | 81.1 | 95.2 | 82.7 | 96.1 |
| MViT-B,16×4(Fan et al., 2021) | - | 16×1×5 | 353 | 78.4 | 93.5 | 82.1 | 95.7 |
| MViT-B,32×3(Fan et al., 2021) | - | 32×1×5 | 850 | 80.2 | 94.4 | 83.4 | 96.3 |
| ViViT-L(Arnab et al., 2021) | IN-21K | 16×3×4 | 17352 | 80.6 | 94.7 | 82.5 | 95.6 |
| ViViT-L(Arnab et al., 2021) | JFT-300M | 16×3×4 | 17352 | 82.8 | 95.3 | 84.3 | 96.2 |
| Swin-T(Liu et al., 2021b) | IN-1K | 32×3×4 | 1056 | 78.8 | 93.6 | - | - |
| Swin-B(Liu et al., 2021b) | IN-1K | 32×3×4 | 3384 | 80.6 | 94.6 | - | - |
| Swin-B(Liu et al., 2021b) | IN-21K | 32×3×4 | 3384 | 82.7 | 95.5 | 84.0 | 96.5 |
| Our UniFormer-S | IN-1K | 16×1×4 | 167 | 80.8 | 94.7 | 82.8 | 95.8 |
| Our UniFormer-B | IN-1K | 16×1×4 | 389 | 82.0 | 95.1 | 84.0 | 96.4 |
| Our UniFormer-B | IN-1K | 32×1×4 | 1036 | 82.9 | 95.4 | 84.8 | 96.7 |
| Our UniFormer-B | IN-1K | 32×3×4 | 3108 | **83.0** | 95.4 | **84.9** | **96.7** |

Table 2: **Comparison with the state-of-the-art on Kinetics-400&600.** Our UniFormer outperforms most of the current methods with much fewer computation cost.

we extend the conditional position encoding (CPE) (Chu et al., 2021) to design our DPE:

$$\text{DPE}(\mathbf{X}_{in}) = \text{DWConv}(\mathbf{X}_{in}), \tag{8}$$

where $\text{DWConv}$ means simple 3D depthwise convolution with zero paddings. Thanks to the shared parameters and locality of convolution, DPE can overcome permutation-invariance and is friendly to arbitrary input lengths. Moreover, it has been proven in CPE that zero paddings help the tokens on the borders be aware of their absolute positions, thus all tokens can progressively encode their absolute spatiotemporal position information via querying their neighbor.

## 3.4 MODEL ARCHITECTURE

We hierarchically stack UniFormer blocks to build up our network for spatiotemporal learning. As shown in Figure 3, our network consists of four stages, the channel numbers of which are 64, 128, 320 and 512 respectively. We provide two model variants depending on the number of UniFormer blocks in these stages: $\{3, 4, 8, 3\}$ for UniFormer-S and $\{5, 8, 20, 7\}$ for UniFormer-B. In the first two stages, we utilize MHRA with local token affinity (Eq. 6) to reduce the short-term spatiotemporal redundancy. The tube size is set to $5 \times 5 \times 5$ and the head number $N$ is equal to the corresponding channel number. In the last two stages, we apply MHRA with global token affinity (Eq. 7) to capture long-term dependency, the head dimension of which is 64. We utilize BN (Ioffe & Szegedy, 2015) for local MHRA and LN (Ba et al., 2016) for global MHRA. The kernel size of DPE is $3 \times 3 \times 3$ ($T \times H \times W$) and the expand ratios of FFN in all layers are 4. We adopt a $3 \times 4 \times 4$ convolution with stride $2 \times 4 \times 4$ before the first stage, which means the spatial and temporal dimensions are both downsampled. Before other stages, we apply $1 \times 2 \times 2$ convolutions with stride $1 \times 2 \times 2$. Finally, the spatiotemporal average pooling and fully connected layer are utilized to output the final predictions.

**Comparison to Convolution+Transformer Network.** The prior works have demonstrate that self-attention can perform convolution (Ramachandran et al., 2019; Cordonnier et al., 2020), but they propose to replace convolution instead of combining them. Recent works have attempted to introduce convolution to vision transformers (Wu et al., 2021; Dai et al., 2021; Gao et al., 2021; Srinivas et al., 2021), but they mainly focus on image recognition, without any spatiotemporal consideration for video understanding. Moreover, the combination is almost straightforward in the prior video transformers, e.g., using transformer as global attention (Wang et al., 2018) or using convolution as patch stem (Liu et al., 2020b). In contrast, our UniFormer tackles both video redundancy and dependency with an insightful unified framework (Table 1). Via local and global token affinity learning, we can achieve a preferable computation-accuracy balance for video classification.

| Method | Pretrain | #Frame | GFLOPs | SSV1 | | SSV2 | |
|---|---|---|---|---|---|---|---|
| | | | | Top-1 | Top-5 | Top-1 | Top-5 |
| TSN(Wang et al., 2016) | IN-1K | 16×1×1 | 66 | 19.9 | 47.3 | 30.0 | 60.5 |
| TSM(Lin et al., 2019) | IN-1K | 16×1×1 | 66 | 47.2 | 77.1 | - | - |
| GST(Luo & Yuille, 2019) | IN-1K | 16×1×1 | 59 | 48.6 | 77.9 | 62.6 | 87.9 |
| MSNet(Kwon et al., 2020) | IN-1K | 16×1×1 | 101 | 52.1 | 82.3 | 64.7 | 89.4 |
| CT-Net(Li et al., 2021a) | IN-1K | 16×1×1 | 75 | 52.5 | 80.9 | 64.5 | 89.3 |
| CT-Net$_{EN}$(Li et al., 2021a) | IN-1K | 8+12+16+24 | 280 | 56.6 | 83.9 | 67.8 | 91.1 |
| TDN(Wang et al., 2020b) | IN-1K | 16×1×1 | 72 | 53.9 | 82.1 | 65.3 | 89.5 |
| TDN$_{EN}$(Wang et al., 2020b) | IN-1K | 8+16 | 198 | 56.8 | 84.1 | 68.2 | 91.6 |
| TimeSformer-HR(Bertasius et al., 2021) | IN-21K | 16×3×1 | 5109 | - | - | 62.5 | - |
| X-ViT(Bulat et al., 2021) | IN-21K | 32×3×1 | 1270 | - | - | 65.4 | 90.7 |
| Mformer-L(Patrick et al., 2021) | K400 | 32×3×1 | 3555 | - | - | 68.1 | 91.2 |
| ViViT-L(Arnab et al., 2021) | K400 | 16×3×4 | 11892 | - | - | 65.4 | 89.8 |
| MViT-B,64×3(Fan et al., 2021) | K400 | 64×1×3 | 1365 | - | - | 67.7 | 90.9 |
| MViT-B-24,32×3(Fan et al., 2021) | K600 | 32×1×3 | 708 | - | - | 68.7 | 91.5 |
| Swin-B(Liu et al., 2021b) | K400 | 32×3×1 | 963 | - | - | 69.6 | 92.7 |
| Our UniFormer-S | K400 | 16×1×1 | 42 | 53.8 | 81.9 | 63.5 | 88.5 |
| Our UniFormer-S | K600 | 16×1×1 | 42 | 54.4 | 81.8 | 65.0 | 89.3 |
| Our UniFormer-S | K400 | 16×3×1 | 125 | 57.2 | 84.9 | 67.7 | 91.4 |
| Our UniFormer-S | K600 | 16×3×1 | 125 | 57.6 | 84.9 | 69.4 | 92.1 |
| Our UniFormer-B | K400 | 16×3×1 | 290 | 59.1 | 86.2 | 70.4 | 92.8 |
| Our UniFormer-B | K600 | 16×3×1 | 290 | 58.8 | 86.5 | 70.2 | **93.0** |
| Our UniFormer-B | K400 | 32×3×1 | 777 | 60.9 | 87.3 | **71.2** | 92.8 |
| Our UniFormer-B | K600 | 32×3×1 | 777 | **61.0** | **87.6** | **71.2** | 92.8 |

Table 3: **Comparison with the state-of-the-art on Something-Something V1&V2.** Our Uni-Former achieves new state-of-the-art performances on both datasets.

## 4 EXPERIMENTS

### 4.1 DATASETS AND EXPERIMENTAL SETUP

We conduct experiments on widely-used Kinetics-400 (Carreira & Zisserman, 2017a) and larger benchmark Kinetics-600 (Carreira et al., 2018). We further verify the transfer learning performance on temporal-related datasets Something-Something V1&V2 (Goyal et al., 2017b). For training, we utilize the dense sampling strategy (Wang et al., 2018) for Kinetics and uniform sampling strategy (Wang et al., 2016) for Something-Something. We adopt the same training recipe as MViT (Fan et al., 2021) by default, but the random horizontal flip is not applied for Something-Something. To reduce the total training cost, we inflate the 2D convolution kernels pre-trained on ImageNet for Kinetics (Carreira & Zisserman, 2017b). More implementation specifics are shown in Appendix C. For testing, we explore the sampling strategies in our experiments. To obtain a preferable computation-accuracy balance, we adopt multi-clip testing for Kinetics and multi-crop testing for Something-Something. All scores are averaged for the final prediction.

### 4.2 COMPARISON TO STATE-OF-THE-ART

**Kinetics-400&600.** Table 2 presents comparisons to the state-of-the-art methods on Kinetics-400 and Kinetics-600. The first part shows the prior works using CNN. Compared with SlowFast (Fe-ichtenhofer et al., 2019), our UniFormer-S$_{16f}$ requires **42×** fewer GFLOPs but obtains 1.0% per-formance gain on both datasets. Even compared with MoViNet (Kondratyuk et al., 2021), which is designed through extensive neural architecture search, our model achieves slightly better results with fewer input frames ($16f×4$ vs. $120f$). The second part lists the recent works based on vi-sion transformers. With only ImageNet-1K pre-training, UniFormer-B$_{16f}$ surpasses most of the other backbones with large dataset pre-training. For example, compared with ViViT-L pre-trained from JFT-300M and Swin-B pre-trained from ImageNet-21K, UniFormer-B$_{32f}$ obtains comparable performance with **16.7×** and **3.3×** fewer computation on both Kinetics-400 and Kinetics-600.

**Something-Something V1&V2.** Results on Something-Something V1&V2 are shown in Table 3. Since these datasets depend on temporal relation modeling, it is difficult for the CNN-based methods to capture long-term dependencies, which leads to their worse results. In contrast, transformer-based backbones are good at processing long sequential data and demonstrate better transfer learning capabilities (Zhou et al., 2021). Our UniFormer pre-trained from Kinetis-600 outperforms all the current methods under the same settings. In fact, our best model achieves the new state-of-the-art

| Unified | Joint | DPE | Type | ImageNet | | | | K400 1×4 | | | |
|---|---|---|---|---|---|---|---|---|---|---|---|
| | | | | GFLOPs | #Param | Top-1 | Top-5 | GFLOPs | #Param | Top-1 | Top-5 |
| ✔ | ✔ | ✔ | **LLGG** | 3.6 | 21.5 | **82.9** | **96.2** | 41.8 | 21.4 | **79.3** | **94.3** |
| ✗ | ✔ | ✔ | LLGG | 3.3 | 21.3 | 82.6 | 96.1 | 41.0 | 21.3 | 78.6 | 93.6 |
| ✔ | ✗ | ✔ | LLGG | 3.6 | 21.5 | 82.9 | 96.2 | 36.8 | 27.7 | 78.7 | 94.1 |
| ✔ | ✔ | ✗ | LLGG | 3.6 | 21.5 | 82.4 | 96.0 | 41.4 | 21.3 | 77.6 | 93.5 |
| ✔ | ✔ | ✔ | **LLLL** | 3.7 | 23.3 | 81.9 | 95.9 | 31.6 | 23.7 | 77.2 | 92.9 |
| ✔ | ✔ | ✔ | **LLLG** | 3.7 | 22.2 | 82.5 | 96.1 | 31.6 | 22.4 | 78.4 | 93.3 |
| ✔ | ✔ | ✔ | **LGGG** | 3.6 | 21.6 | 82.7 | 96.1 | 39.0 | 21.4 | 79.0 | 94.1 |
| ✔ | ✔ | ✔ | **GGGG** | 3.7 | 20.1 | 82.1 | 95.9 | 72.0 | 19.8 | 75.3 | 92.4 |

(a) **Structure design.** All models are trained for 50 epochs on Kinetics-400. To guarantee the parameters and computation of all the models are similar, when modifying the stage types, we modify the stage numbers and reduce the computation of self-attention as MViT (Fan et al., 2021) for LGGG and GGGG.

| Size | K400 1×4 | |
|---|---|---|
| | GFLOPs | Top1 |
| 3 | 41.0 | 79.0 |
| 5 | 41.8 | **79.3** |
| 7 | 43.6 | 79.1 |
| 9 | 46.6 | 78.9 |

(b) **Tube size.** Our network is basically robust to the tube size.

| Type | Joint | GFLOPs | Pretrain | SSV1 Top-1 |
|---|---|---|---|---|
| LLLL | ✔ | 26.1 | ImageNet | 49.2 |
| | | | K400 | 49.2(+0.0) |
| LLGG | ✗ | 36.8 | ImageNet | 51.9 |
| | | | K400 | 51.8(−0.1) |
| LLGG | ✔ | 41.8 | ImageNet | 52.0 |
| | | | K400 | **53.8**(+1.8) |

(c) **Transfer learning.** Jointly manner performs better when pre-training from larger dataset.

| Model | Sampling Method | K400 Top-1 | |
|---|---|---|---|
| | | 1×1 | 1×4 |
| Small | 16×4 | 76.2 | **80.8** |
| | 16×8 | **78.4** | 80.7 |
| Base | 16×4 | 78.1 | **82.0** |
| | 16×8 | **79.3** | 81.7 |
| Small | 32×2 | 77.3 | 81.2 |
| | 32×4 | **79.8** | **82.0** |

(d) **Sampling method.**

Table 4: **Ablation studies.** 'Unified' means whether to use our local MHRA (✗ means to use MobileNet block). 'Joint' means whether to use joint attention. 'L'/'G' refers to local/global MHRA.

results: **61.0%** top-1 accuracy on Something-Something V1 (**4.2%** higher than TDN$_{EN}$) (Wang et al., 2020b) and **71.2%** top-1 accuracy on Something-Something V2 (**1.6%** higher than Swin-B (Liu et al., 2021b)). Such results verify the capability of spatiotemporal learning for UniFormer.

## 4.3 ABLATION STUDIES

**UniFormer vs. Convolution: Does transformer-style FFN help?** As mentioned in Section 3.2, our UniFormer block in the shallow layers can be interpreted as a transformer-style spatiotemporal MobileNet block (Tran et al., 2019) with extra FFN. Hence, we first investigate its effectiveness by replacing our UniFormer blocks in shallow layers with MobileNet blocks (the expand ratios are set to 3 for similar parameters). As expected, our default UniFormer outperforms such spatiotemporal MobileNet block in Table 4a. It shows that, FFN in our UniFormer can further mix token context at each spatiotemporal position to boost classification accuracy.

**UniFormer vs. Transformer: Is joint or divided spatiotemporal attention better?** As discussed in Section 3.2, our UniFormer block in the deep layers can be interpreted as a transformer block, but our attention is jointly learned in a spatiotemporal manner, instead of dividing spatial and temporal attention (Bertasius et al., 2021; Arnab et al., 2021). As shown in Table 4a, the joint version is more powerful than the separate one, showing that joint spatiotemporal attention can learn more discriminative video representations. What's more, the joint attention is more friendly to transfer learning with pre-training. As shown in Table 4c, when the model is gradually pre-trained from ImageNet to Kinetics-400, the performance of our UniFormer becomes better. Such distinct characteristic is not observed in the pure local MHRA structure (LLLL) and the splitting version. It demonstrates that the joint learning manner is preferable for video representation learning.

**Does dynamic position embedding matter to UniFormer?** With dynamic position embedding, our UniFormer improve the top-1 accuracy by 0.5% and 1.7% on ImageNet and Kinetics-400. It shows that via encoding the position information, our DPE can maintain spatiotemporal order, contributing to better spatiotemporal representation learning.

**How much does local MHRA help?** Since our UniFormer is equipped with local and global token affinity respectively in the shallow and deep layers, we investigate the configuration of our network stage by stage. As shown in Table 4a, when we only use local MHRA (LLLL), the computation cost will be light. However, the accuracy is largely dropped, since the network lacks the capacity of learning long-term dependency without global MHRA. When we gradually replace local MHRA with global MHRA, the accuracy becomes better as expected. However, the accuracy is dramatically dropped with a heavy computation load when all the layers apply global MHRA (GGGG). It is

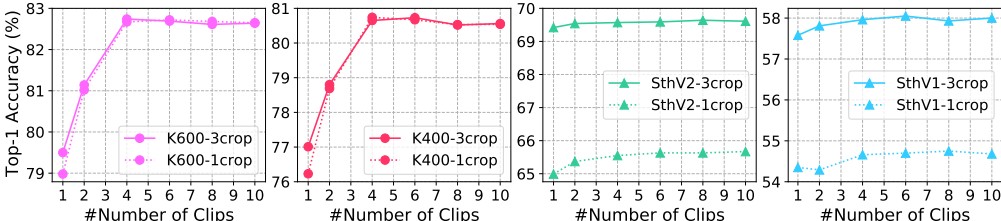

Figure 4: **Multi-clip/crop testing comparison on different datasets.** Multi-clip testing is better for Kinetics and multi-crop testing is better for Something-Something.

mainly because that, without local MHRA, the network lacks the capacity of extracting detailed video representations, leading to severe model overfitting with redundant spatiotemporal attention. In our experiments, we choose local MHRA and global MHRA in the first two stages and the last two stages respectively, in order to achieve a preferable computation-accuracy balance.

**Is our UniFormer more transferable?** We further verify the transfer learning ability of our Uni-Former in Table 4c. All models share the same stage numbers but the stage types are different. Compared with pre-training from ImgeNet, pre-training from Kinetics-400 will further improve the top-1 accuracy by 1.8%. However, such distinct characteristic is not observed in the pure local MHRA structure and UniFormer with divided spatiotemporal attention. It demonstrates that the joint learning manner is preferable for transfer learning.

**Empirical investigation on model parameters.** We further evaluate the robustness of our Uni-Former network to several important model parameters. **(1) size of local tube:** In our local token affinity (Eq. 6), we aggregate spatiotemporal context from a small local tube. Hence, we investigate the influence of this tube by changing its 3D size (Table 4b). Our network is robust to the tube size. We choose 5×5×5 for better accuracy. **(2) sampling method:** We explore the vital sampling method shown in Table 4d. For training, 16×4 means that we sample 16 frames with frame stride 4. For testing, 4×1 means four-clip testing. As expected, sparser sampling method achieves a higher single-clip result. For multi-clip testing, dense sampling is slightly better when sampling a few frames. However, when sampling more frames, sparse sampling is obviously better. **(3) testing strategy:** We evaluate our network with different numbers of clips and crops for the validation videos. As shown in Figure 4, since Kinetics is a scene-related dataset and trained with dense sampling, multi-clip testing is preferable to cover more frames for boosting performance. Alternatively, Something-Something is a temporal-related dataset and trained with uniform sampling, so multi-crop testing is better for capturing the discriminative motion for boosting performance.

## 4.4 VISUALIZATION

To further verify the effectiveness of UniFormer, we conduct some visualizations of different structures (see Appendix D). In Figure 5, We apply Grad-CAM (Selvaraju et al., 2019) to show the areas of the greatest concern in the last layer. It reveals that GGGG struggles to focus on the key object, i.e., the skateboard and the football, as it blindly compares the similarity of all tokens in all layers. Alternatively, LLLL only performs local aggregation. Hence, its attention tends to be coarse and inaccurate without a global view. Different from both cases, our UniFormer with LLGG can cooperatively learn local and global contexts in a joint manner. As a result, it can effectively capture the most discriminative information, by paying precise attention to the skateboard and the football. In Figure 6, we present the accuracies of different structures on Kinetics-400 (Carreira & Zisserman, 2017a). It shows that LLGG outperforms other structures in most categories, which demonstrates that our UniFormer takes advantage of both 3D convolution and spatiotemporal self-attention.

## 5 CONCLUSION

In this paper, we propose a novel UniFormer, which can effectively unify 3D convolution and spatiotemporal self-attention in a concise transformer format to overcome video redundancy and dependency. We adopt local MHRA in shallow layers to largely reduce computation burden and global MHRA in deep layers to learn global token relation. Extensive experiments demonstrate that our UniFormer achieves a preferable balance between accuracy and efficiency on popular video benchmarks, Kinetics-400/600 and Something-Something V1/V2.

## ACKNOWLEDGEMENT

This work is partially supported bythe National Natural Science Foundation of China (61876176,U1813218), Guangdong NSF Project (No. 2020B1515120085), the Shenzhen Research Program(RCJC20200714114557087), the Shanghai Committee of Science and Technology, China (Grant No. 21DZ1100100).

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

## A    MORE DETAILS ABOUT LOCAL MHRA

For local MHRA, it is vital to determine the neighbor tokens. Considering any token $\mathbf{X}_k$ ($k \in [0, L-1]$), we can calculate its index $(t_k, h_k, w_k)$ as follows:

$$t_k = \lfloor \frac{k}{H \times W} \rfloor, \tag{9}$$

$$h_k = \lfloor \frac{k - t_k \times H \times W}{W} \rfloor, \tag{10}$$

$$w_k = (k - t_k \times H \times W) \, mod \, W. \tag{11}$$

Therefore, for an anchor token $\mathbf{X}_i$, any of its neighbor tokens $\mathbf{X}_j$ in $\Omega_i^{t \times h \times w}$ should satisfy

$$|t_i - t_j| \leq \frac{t}{2}, \tag{12}$$

$$|h_i - h_j| \leq \frac{h}{2}, \tag{13}$$

$$|w_i - w_j| \leq \frac{w}{2}. \tag{14}$$

Thus the local spatiotemporal affinity in Eq. 6 can be calculated as follows:

$$\mathrm{A}_n^{local}(\mathbf{X}_i, \mathbf{X}_j) = a^{i-j} \tag{15}$$

$$= a[t_i - t_j, h_i - h_j, w_i - w_j]. \tag{16}$$

For other tokens not in $\Omega_i^{t \times h \times w}$, $\mathrm{A}_n^{local}(\mathbf{X}_i, \mathbf{X}_j) = 0$.

## B    MORE DETAILS ABOUT FFN.

We adopt the standard FFN (Eq. 3) in vision transformers (Dosovitskiy et al., 2021),

$$\mathbf{Z}'' = \mathrm{Linear}_2 \left( \mathrm{GELU} \left( \mathrm{Linear}_1 \left( \mathbf{Z}' \right) \right) \right), \tag{17}$$

where GELU is a non-linear function. The channel number will be first expanded by ratio 4 and then reduced. All token representations will be enhanced after performing FFN.

## C    ADDITIONAL IMPLEMENTATION DETAILS

**Architecture details.** As in ViT (Dosovitskiy et al., 2021), we adopt the pre-normalization configuration (Wang et al., 2019) that applies norm layer at the beginning of the residual function (He et al., 2016). Differently, we utilize BN (Ioffe & Szegedy, 2015) for local MHRA and LN (Ba et al., 2016) for global MHRA. Moreover, we add an extra layer normalization in the downsampling layers.

**Training details.** We adopt AdamW (Loshchilov & Hutter, 2017a) optimizer with cosine learning rate schedule (Loshchilov & Hutter, 2017b) to train the entire network. The first 5 or 10 epochs are used for warm-up (Goyal et al., 2017a) to overcome early optimization difficulty. For UniFormer-S, the warmup epoch, total epoch, stochastic depth rate, weight decay are set to 10, 110, 0.1 and 0.05 respectively for Kinetics and 5, 50, 0.3 and 0.05 respectively for Something-Something. For UniFormer-B, all the hyper-parameters are the same unless the stochastic depth rates are doubled. We linearly scale the base learning rates according to the batch size, which are $1e^{-4} \times \frac{batchsize}{32}$ and $2e^{-4} \times \frac{batchsize}{32}$ for Kinetics and Something-Something.

## D    VISUALIZATION

We choose three structures used in our experiments (Table 4a) to make comparisons: LLGG, LLLL and GGGG. The stage numbers of them are $\{3, 4, 8, 3\}$, $\{3, 5, 10, 4\}$ and $\{2, 2, 7, 3\}$ respectively.

In Figure 5, we conduct attention visualization of different structures. Part1 shows the input videos selected from Kinetics-400 (Carreira & Zisserman, 2017a). In part2, we use Grad-CAM (Selvaraju

| Method | Sampling stride | #Frame | GFLOPs | #Param | K400 | | K600 | |
|---|---|---|---|---|---|---|---|---|
| | | | | | Top-1 | Top-5 | Top-1 | Top-5 |
| UniFormer-S | 4 | 16×1×1 | 41.8 | 21.4 | 76.2 | 92.2 | 79.0 | 93.6 |
| | | 16×1×4 | 167.2 | 21.4 | 80.8 | 94.7 | 82.8 | 95.8 |
| | 8 | 16×1×1 | 41.8 | 21.4 | 78.4 | 92.9 | 80.8 | 94.7 |
| | | 16×1×4 | 167.2 | 21.4 | 80.8 | 94.4 | 82.7 | 95.7 |
| | 2 | 32×1×1 | 109.6 | 21.4 | 77.3 | 92.4 | - | - |
| | | 32×1×4 | 438.4 | 21.4 | 81.2 | 94.7 | - | - |
| | 4 | 32×1×1 | 109.6 | 21.4 | 79.8 | 93.4 | - | - |
| | | 32×1×4 | 438.4 | 21.4 | 82.0 | 95.1 | - | - |
| UniFormer-B | 4 | 16×1×1 | 96.7 | 49.8 | 78.1 | 92.8 | 80.3 | 94.5 |
| | | 16×1×4 | 386.8 | 49.8 | 82.0 | 95.1 | 84.0 | 96.4 |
| | 8 | 16×1×1 | 96.7 | 49.8 | 79.3 | 93.4 | 81.7 | 95.0 |
| | | 16×1×4 | 386.8 | 49.8 | 81.7 | 94.8 | 83.4 | 96.0 |
| | 4 | 32×1×1 | 259 | 49.8 | 80.9 | 94.0 | 82.7 | 95.7 |
| | | 32×1×4 | 1036 | 49.8 | **82.9** | **95.4** | **84.8** | **96.7** |

Table 5: More results on Kinetics-400&600.

et al., 2019) to generate the corresponding attention in the last layer. Since GGGG blindly compares the similarity of all tokens in all, it struggles to focus on the key object, i.e., the skateboard and the football. Alternatively, LLLL only performs local aggregation without a global view, leading to coarse and inaccurate attention. Different from both cases, our UniFormer with LLGG can cooperatively learn local and global contexts in a joint manner. As a result, it can effectively capture the most discriminative information, by paying precise attention to the skateboard and the football.

Additionally, in Figure 6, we show the top-1 accuracies of different structures on Kinetics-400 (Carreira & Zisserman, 2017a). It demonstrates that GGGG surpasses the other two structures in most categories. Furthermore, we analyze the prediction results of several categories in Figure 7. It shows that *gargling* is often misjudged as *brushing teeth*, while *swing dancing* is often misjudged as other types of dancing. We argue that these categories are easier to be discriminated against based on the spatial details. For example, toothbrush often exists in *brushing teeth* but not in *gargling*, and the people's poses are different in different dancing. Therefore, LLLL performs better than GGGG in these categories thanks to the capacity of encoding detailed spatiotemporal features. What's more, *playing guitar* and *strumming guitar* are difficult to be classified, since their spatial contents are almost the same. They require the long-range dependency between objects, e.g., the interaction between the people's hand and the guitar, thus GGGG does better. More importantly, our UniFormer with LLGG is competitive with the other two methods in these categories, which means it takes advantage of both 3D convolution and spatiotemporal self-attention.

# E    ADDITIONAL RESULTS

## E.1    MORE RESULTS ON KINETICS

Table 5 shows more results on Kinetics-400 (Carreira & Zisserman, 2017a) and Kinetics-600 (Carreira et al., 2018). The trends of the results on both datasets are similar. When sampling with a large frame stride, the corresponding single-clip testing result will be better. It is mainly because sparser sampling covers a larger time range. For multi-clip testing, sampling with frame stride 4 always performs better, thus we adopt frame stride 4 by default.

## E.2    MORE RESULTS ON SOMETHING-SOMETHING

Table 6 presents more results on Something-Something V1&V2 (Goyal et al., 2017b). For UniFormer-S, pre-training with Kinetics-600 is better than pre-training with Kinetics-400, improving the top-1 accuracy by approximately 1.5%. However, for UniFormer-B, the improvement is not obvious. We claim that the small model is difficult to fit, thus larger dataset pre-training can help it fit better. Besides, UniFormer-B with 16 frames performs better than UniFormer-S with 32 frames.

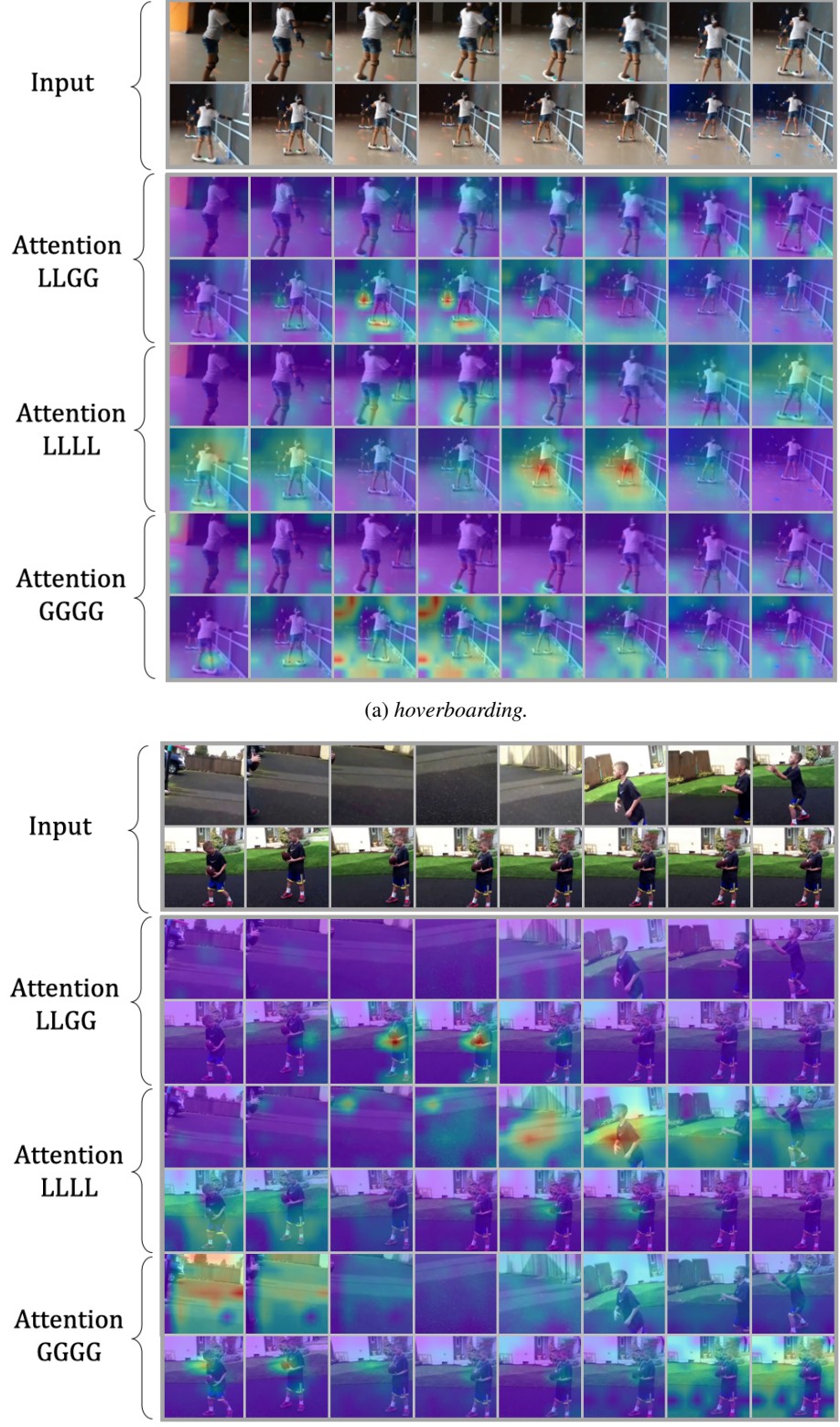

(a) *hoverboarding.*

(b) *passing American football (not in game).*

Figure 5: Attention visualization of different structures. Videos are chosen from Kinetics-400 (Carreira & Zisserman, 2017a).

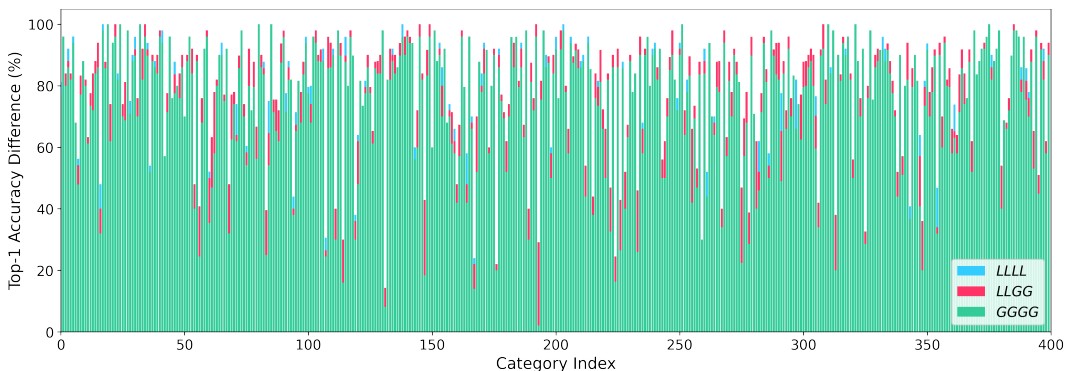

Figure 6: Accuracy of different structures on Kinetics-400.

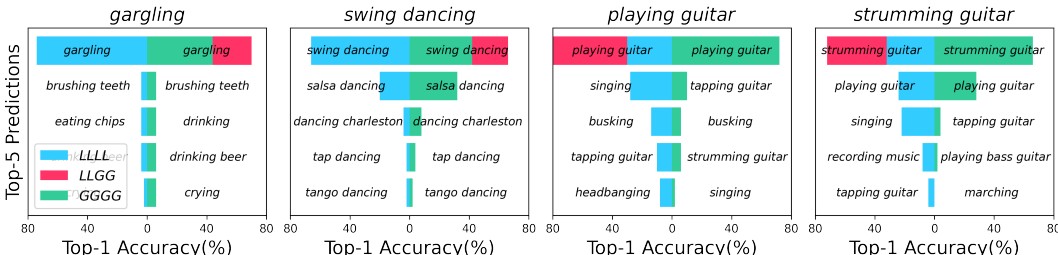

Figure 7: Prediction comparisons of different structures.

| Method | Pretrain | #Frame | GFLOPs | #Param | SSV1 | | SSV2 | |
|--------|----------|--------|--------|--------|------|------|------|------|
| | | | | | Top-1 | Top-5 | Top-1 | Top-5 |
| UniFormer-S | K400 | 16×1×1 | 41.8 | 21.3 | 53.8 | 81.9 | 63.5 | 88.5 |
| | | 16×3×1 | 125.4 | 21.3 | 57.2 | 84.9 | 67.7 | 91.4 |
| | | 16×3×2 | 250.8 | 21.3 | 57.3 | 85.1 | 68.1 | 91.7 |
| | K600 | 16×1×1 | 41.8 | 21.3 | 54.4 | 81.8 | 65.0 | 89.3 |
| | | 16×3×1 | 125.4 | 21.3 | 57.6 | 84.9 | 69.4 | 92.1 |
| | | 16×3×2 | 250.8 | 21.3 | 57.8 | 84.9 | 69.5 | 92.2 |
| | K400 | 32×1×1 | 109.6 | 21.3 | 55.8 | 83.6 | 64.9 | 89.2 |
| | | 32×3×1 | 328.8 | 21.3 | 58.8 | 86.4 | 69.0 | 91.7 |
| | | 32×3×2 | 657.6 | 21.3 | 58.9 | 86.6 | 69.2 | 91.8 |
| | K600 | 32×1×1 | 109.6 | 21.3 | 56.9 | 83.8 | 66.4 | 90.2 |
| | | 32×3×1 | 328.8 | 21.3 | 59.9 | 86.2 | 70.4 | **93.1** |
| | | 32×3×2 | 657.6 | 21.3 | 59.9 | 86.3 | 70.5 | 92.9 |
| UniFormer-B | K400 | 16×1×1 | 96.7 | 49.7 | 55.4 | 82.9 | 65.8 | 89.9 |
| | | 16×3×1 | 290.1 | 49.7 | 59.1 | 86.2 | 70.4 | 92.8 |
| | | 16×3×2 | 580.2 | 49.7 | 59.3 | 86.4 | 70.7 | 92.9 |
| | K600 | 16×1×1 | 96.7 | 49.7 | 55.7 | 83.3 | 66.1 | 90.0 |
| | | 16×3×1 | 290.1 | 49.7 | 58.8 | 86.5 | 70.2 | 93.0 |
| | | 16×3×2 | 580.2 | 49.7 | 59.1 | 86.5 | 70.7 | 92.9 |
| | K400 | 32×1×1 | 259 | 49.7 | 58.1 | 84.9 | 67.2 | 90.2 |
| | | 32×3×1 | 777 | 49.7 | 60.9 | 87.3 | 71.2 | 92.8 |
| | | 32×3×2 | 1554 | 49.7 | 61.0 | 87.3 | **71.4** | 92.8 |
| | K600 | 32×1×1 | 259 | 49.7 | 58.0 | 84.9 | 67.5 | 90.2 |
| | | 32×3×1 | 777 | 49.7 | 61.0 | 87.6 | 71.2 | 92.8 |
| | | 32×3×2 | 1554 | 49.7 | **61.2** | **87.6** | 71.3 | 92.8 |

Table 6: More results on Something-Something V1&V2.

### E.3 Comparsion to state-of-the-art on ImageNet

Table 7 compares our method with the state-of-the-art ImageNet (Deng et al., 2009). We design four model variants as follows:

- UniFormer-S: channel numbers=$\{64, 128, 320, 512\}$, stage numbers=$\{3, 4, 8, 3\}$
- UniFormer-S†: channel numbers=$\{64, 128, 320, 512\}$, stage numbers=$\{3, 5, 9, 3\}$
- UniFormer-B: channel numbers=$\{64, 128, 320, 512\}$, stage numbers=$\{5, 8, 20, 7\}$
- UniFormer-L: channel numbers=$\{128, 192, 448, 640\}$, stage numbers=$\{5, 10, 24, 7\}$

All the other model parameters are the same as we mention in Section 3.4. For UniFormer-S†, we adopt overlapped convolutional patch embedding. All the training hyper-parameters are the same as DeiT (Touvron et al., 2021a) by defaults. When training our models with Token Labeling, we follow the settings used in LV-ViT (Jiang et al., 2021). It shows that our models outperform other methods with similar parameters/FLOPs on ImageNet, especially when training with Token Labeling. Moreover, our model surpasses those models combining CNN with Transformer, e.g., CvT (Wu et al., 2021) and CoAtNet (Dai et al., 2021), which reflects our UniFormer can unify convolution and self-attention better for preferable accuracy-computation balance.

| Method | Architecture | #Param | GFLOPs | Train Size | Test Size | ImageNet Top-1 |
|---|---|---|---|---|---|---|
| RegNetY-4G (Radosavovic et al., 2020) | CNN | 21 | 4.0 | 224 | 224 | 80.0 |
| EffcientNet-B5 (Tan & Le, 2019) | CNN | 30 | 9.9 | 456 | 456 | 83.6 |
| EfficientNetV2-S (Tan & Le, 2021) | CNN | 22 | 8.5 | 384 | 384 | 83.9 |
| DeiT-S (Touvron et al., 2021a) | Trans | 22 | 4.6 | 224 | 224 | 79.9 |
| PVT-S (Wang et al., 2021) | Trans | 25 | 3.8 | 224 | 224 | 79.8 |
| T2T-14 (Yuan et al., 2021) | Trans | 22 | 5.2 | 224 | 224 | 80.7 |
| Swin-T (Liu et al., 2021a) | Trans | 29 | 4.5 | 224 | 224 | 81.3 |
| CSwin-T ↑384 (Dong et al., 2021) | Trans | 23 | 14.0 | 224 | 384 | 84.3 |
| LV-ViT-S (Jiang et al., 2021) | Trans | 26 | 6.6 | 224 | 224 | 83.3 |
| LV-ViT-S ↑384 (Jiang et al., 2021) | Trans | 26 | 22.2 | 224 | 384 | 84.4 |
| CvT-13 (Wu et al., 2021) | CNN+Trans | 20 | 4.5 | 224 | 224 | 81.6 |
| CvT-13 ↑384 (Wu et al., 2021) | CNN+Trans | 20 | 16.3 | 224 | 384 | 83.0 |
| CoAtNet-0 (Dai et al., 2021) | CNN+Trans | 25 | 4.2 | 224 | 224 | 81.6 |
| CoAtNet-0 ↑384 (Dai et al., 2021) | CNN+Trans | 20 | 13.4 | 224 | 384 | 83.9 |
| Container (Gao et al., 2021) | CNN+Trans | 22 | 8.1 | 224 | 224 | 82.7 |
| UniFormer-S | CNN+Trans | 22 | 3.6 | 224 | 224 | 82.9 |
| UniFormer-S+TL | CNN+Trans | 22 | 3.6 | 224 | 224 | 83.4 |
| UniFormer-S+TL ↑384 | CNN+Trans | 22 | 11.9 | 224 | 384 | 84.6 |
| UniFormer-S† | CNN+Trans | 24 | 4.2 | 224 | 224 | 83.4 |
| UniFormer-S†+TL | CNN+Trans | 24 | 4.2 | 224 | 224 | 83.9 |
| UniFormer-S†+TL ↑384 | CNN+Trans | 24 | 13.7 | 224 | 384 | **84.9** |
| RegNetY-8G (Radosavovic et al., 2020) | CNN | 39 | 8.0 | 224 | 224 | 81.7 |
| EffcientNet-B7 (Tan & Le, 2019) | CNN | 66 | 39.2 | 600 | 600 | 84.3 |
| EfficientNetV2-M (Tan & Le, 2021) | CNN | 54 | 25.0 | 480 | 480 | 85.1 |
| PVT-L (Wang et al., 2021) | Trans | 61 | 9.8 | 224 | 224 | 81.7 |
| T2T-24 (Yuan et al., 2021) | Trans | 64 | 13.2 | 224 | 224 | 82.2 |
| Swin-S (Liu et al., 2021a) | Trans | 50 | 8.7 | 224 | 224 | 83.0 |
| CSwin-S ↑384 (Dong et al., 2021) | Trans | 35 | 22.0 | 224 | 384 | 85.0 |
| LV-ViT-M (Jiang et al., 2021) | Trans | 56 | 16.0 | 224 | 224 | 84.1 |
| LV-ViT-M ↑384 (Jiang et al., 2021) | Trans | 56 | 42.2 | 224 | 384 | 85.4 |
| CvT-21 (Wu et al., 2021) | CNN+Trans | 32 | 7.1 | 224 | 224 | 82.5 |
| CvT-21 ↑384 (Wu et al., 2021) | CNN+Trans | 32 | 24.9 | 224 | 384 | 83.3 |
| CoAtNet-1 (Dai et al., 2021) | CNN+Trans | 42 | 8.4 | 224 | 224 | 83.3 |
| CoAtNet-1 ↑384 (Dai et al., 2021) | CNN+Trans | 42 | 27.4 | 224 | 384 | 85.1 |
| UniFormer-B | CNN+Trans | 50 | 8.3 | 224 | 224 | 83.9 |
| UniFormer-B+TL | CNN+Trans | 50 | 8.3 | 224 | 224 | 85.1 |
| UniFormer-B+TL ↑384 | CNN+Trans | 50 | 27.2 | 224 | 384 | **86.0** |
| RegNetY-16G (Radosavovic et al., 2020) | CNN | 84 | 16.0 | 224 | 224 | 82.9 |
| EfficientNetV2-L (Tan & Le, 2021) | CNN | 121 | 53 | 480 | 480 | 85.7 |
| NFNet-F4 (Brock et al., 2021) | CNN | 316 | 215.3 | 384 | 512 | 85.9 |
| NFNet-F5 (Brock et al., 2021) | CNN | 377 | 289.8 | 416 | 544 | 86.0 |
| DeiT-B Touvron et al. (2021a) | Trans | 86 | 17.5 | 224 | 224 | 81.8 |
| Swin-B (Liu et al., 2021a) | Trans | 88 | 15.4 | 224 | 224 | 83.3 |
| CSwin-B ↑384 (Dong et al., 2021) | Trans | 78 | 47.0 | 224 | 384 | 85.4 |
| LV-ViT-L (Jiang et al., 2021) | Trans | 150 | 59.0 | 288 | 288 | 85.3 |
| LV-ViT-L ↑448 (Jiang et al., 2021) | Trans | 150 | 157.2 | 288 | 448 | 85.9 |
| CaiT-S48 ↑384 (Touvron et al., 2021b) | Trans | 89 | 63.8 | 224 | 384 | 85.1 |
| CaiT-M36 ↑448ϒ (Touvron et al., 2021b) | Trans | 271 | 247.8 | 224 | 448 | 86.3 |
| BoTNet-T7 (Srinivas et al., 2021) | CNN+Trans | 79 | 19.3 | 256 | 256 | 84.2 |
| BoTNet-T7 ↑384 (Srinivas et al., 2021) | CNN+Trans | 79 | 45.8 | 256 | 384 | 84.7 |
| CoAtNet-3 (Dai et al., 2021) | CNN+Trans | 168 | 34.7 | 224 | 224 | 84.5 |
| CoAtNet-3 ↑384 (Dai et al., 2021) | CNN+Trans | 168 | 107.4 | 224 | 384 | 85.8 |
| UniFormer-L+TL | CNN+Trans | 100 | 12.6 | 224 | 224 | 85.6 |
| UniFormer-L+TL ↑384 | CNN+Trans | 100 | 39.2 | 224 | 384 | **86.3** |

Table 7: Comparison with the state-of-the-art on ImageNet. 'Train Size' and 'Test Size' refer to resolutions used in training and fine-tuning respectively. 'TL' means token labeling proposed in LV-ViT (Jiang et al., 2021). We group the models based on their parameters.

