# OpenReview forum: "UniFormer: Unified Transformer for Efficient Spatial-Temporal Representation Learning"
_ICLR.cc/2022/Conference — ICLR 2022 Poster_

### Official Review · Reviewer_rXsf · 2021-10-26

**Correctness:** 3
**Technical Novelty And Significance:** 2
**Empirical Novelty And Significance:** 4
**Recommendation:** 8
**Confidence:** 4

**Main Review:**

Strengths:
+ The authors tackle an important problem of interest to many researchers in the video community.
+ Very strong results on multiple action recognition benchmarks.
+ Impressive efficiency considering how good the action recognition accuracy is.
+ Extensive ablation experiments.

Weaknesses:
- The paper is poorly written and is quite difficult to understand.
- The technical novelty of the paper is limited.

In my view, these two weaknesses are connected as I will discuss next. As discussed in Section 3.4, the proposed local UniFormer block is very similar to depthwise 3D convolutions. I appreciate that the authors included this section in their draft, as it makes it easier to understand their approach. However, before this section, the authors spend several paragraphs "selling" this local UniFormer block as a novel transformer block design. I think that's a bit misleading to the readers, and it also makes the paper much more difficult to read (in my view). At this point, most readers are probably familiar with depthwise 3D convolutions. Therefore, in my view, from the very beginning, the authors should describe their local block design by connecting it closely to the concept of depthwise 3D convolutions. Instead, section 3.2 is presented as a novel transformer block with many important details only included in the Appendix. This makes the draft convoluted and harder to read than it should be. Furthermore, the missing technical details wouldn't even be necessary to include if the proposed block was formulated as a depthwise 3D convolution extension.

Connected to my point above, I don't know if the authors are using this strange formulation of their local block to increase the technical novelty of their proposed approach. However, considering the similarities with a 3D MobileNet block, I think the technical novelty of the proposed approach is quite limited. The proposed model is essentially a combination of 3D CNNs and Vision Transformers (with some bells and whistles). Personally, I wouldn't even have an issue with this because, in my view, understanding how to combine 3D convolutions and spatiotemporal self-attention is an important research question. However, I have a problem that the paper is "being sold" as a novel transformer-based architecture, as opposed to an extensive and valuable empirical study of combining 3D convolutional, and self-attention blocks. In a way, this paper reminds me a lot of the R(2+1)D paper "A Closer Look at Spatiotemporal Convolutions for Action Recognition", and I think it should be written in a similar style as this paper (i.e., empirical paper as opposed to claiming a novel architecture).  The authors might argue that the proposed local block is different than a depthwise 3D MobileNet block by pointing to the results in Table 4. However, in my view, those results largely support my claims, i.e., the gain of using the proposed block over the standard MobileNet block is marginal. Furthermore, considering my points above, I'm also not sure if UniFormer is the best name for the proposed approach. It's vague and it doesn't indicate in any way that the unification is between 3D convolutions and self-attention (as opposed to different modalities, different video tasks, or something else that's much broader).

Lastly, I also wanted to mention that the section on DPE is not very informative. The authors simply mention that they "extend the conditional position encoding (CPE) (Chu et al., 2021), utilizing a simple 3D depthwise convolution with zero paddings as dynamic position embedding (DPE)". It would be useful to add more details on this for those readers who are not familiar with CPE.

There are also typos, and grammar mistakes in the draft (a few of them listed below):
- "Sprots" -> "Sports" in Table 2
- Wrong ViViT citation in Table 3
- "Spatial-Temporal" -> "Spatiotemporal"

I'd be interested to hear how the authors plan to address my concerns in the camera-ready version of the draft.

**Summary Of The Paper:**

The paper proposes a UniFormer architecture, which combines 3D convolutions and spatiotemporal self-attention for efficient and effective video classification performance. Specifically, the authors show that using local 3D convolution-like layers in the early stages of the network and global self-attention layers in the later stages of the network provides a good tradeoff between efficiency and accuracy. The proposed method reports state-of-the-art results on several major action recognition benchmarks.

**Summary Of The Review:**

Despite the shortcomings listed above, I'm still largely in favor of accepting this paper. It provides a very valuable empirical study of combining 3D convolution-based operators with self-attention. Furthermore, the authors obtain very impressive results at a relatively small computational cost (especially compared to prior video transformer papers). The presented ablation studies would also be highly beneficial for the research community.

---

> ### Author Response · Authors · 2021-11-21
> **Response to Reviewer rXsf**
>
> Thanks for your constructive comments.
> We provide our feedbacks and modify our paper as follows.
> ------
> **Q1: The proposed local UniFormer block is similar to depthwise 3D convolutions. The authors show this local block as a novel transformer block design. I think that's a bit misleading.**
>
> **A1:**
> We appreciate your detailed and insightful comment.
> For clarification,
> we have added explicit descriptions to connect our local design with the 3D convolution concept in Section 3.2 of our revision (Page 4).
> Indeed,
> we instantiate token affinity as 3D MobileNet in the shallow layers.
> However,
> this is the partial design of our local UniFormer.
> Since our initial motivation is to design a computational-efficient transformer,
> we also contain DPE and FFN in our local UniFormer.
> This has not been explored in the previous 3D convolutional blocks.
> Hence,
> our local block is in a transformer format.
> ------
> **Q2:
> Considering the similarities with a 3D MobileNet block,
> I think the technical novelty is limited.
> Understanding how to combine 3D convolutions and spatiotemporal self-attention is an important research question.
> I think it should be written in a similar style as an empirical paper as opposed to claiming a novel architecture.**
>
> **A2:**
> Thanks for your valuable suggestions.
> Firstly,
> we would like to briefly discuss the novel designs of our UniFormer.
>
> - As mentioned in the paper,
> our initial motivation is to develop an efficient video classification model via a concise transformer format.
> Without this view,
> it is not trivial to design our local and global blocks,
> by an empirical combination of 3D convolution and transformer.
> For example,
> connecting self-attention with 3D convolution is only our partial design in the local block.
> The local block also contains important dynamical position embedding (DPE) and FFN.
> Besides,
> our global block is also a transformer but with DPE instantiated by 3D convolution.
> Such designs have not been explored in the previous 3D CNNs and video transformers.
> - As shown in Table 4(a) (Page 8),
> without our DPE,
> the top-1 accuracy drops by **1.7%** (79.3% vs. 77.6%).
> It demonstrates that,
> DPE is an important unit in our local UniFormer block,
> which does not exist in the standard 3D MobileNet block.
>
> Due to the limited rebuttal period,
> we are extending this work as a journal paper,
> where we will further investigate the relation between 3D convolution and self-attention,
> and make the whole work an empirical paper with extensive comparisons.
> Thanks again for your detailed comments.
> ------
> **Q3: The name UniFormer doesn't indicate in any way that the unification is between 3D convolutions and self-attention.**
>
> **A3:**
> Thanks for your suggestion.
> To avoid ambiguity,
> we explicitly claim the name UniFormer in the introduction of our revision (Page 2).
> It aims to effectively unify 3D convolution and spatiotemporal self-attention in a concise transformer format.
> ------
> **Q4: It would be useful to add more details on DPE for those readers who are not familiar with CPE.**
>
> **A4:**
> Thanks for your constructive comment.
> We have added another subsection (Section 3.3 of our revision, Page 5) to further explain our DPE design.
> Since absolute position embedding is hard to handle different input sizes,
> we extend condition position embedding (CPE) [1] to design our DPE.
> It is instantiated as 3D depth-wise convolution with zero paddings,
> thus it can overcome permutation-invariance and is friendly to arbitrary input lengths.
> Moreover,
> it has been demonstrated in CPE that zero paddings can help to encode token position information progressively,
> which is necessary for video understanding.
> ------
> **Q5: Typos and grammar mistakes in the draft.**
>
> **A5:**
> Thanks for this comment.
> We have corrected them in our revision.
>
> ------
> **Reference**
>
> [1] Chu X, Tian Z, Zhang B, et al. Conditional positional encodings for vision transformers[J]. arXiv preprint arXiv:2102.10882, 2021.

---

> > ### Comment · Reviewer_rXsf · 2021-11-29
> > **Post-Rebuttal Comments**
> >
> > I have read the rebuttal, and it addresses most of my initial concerns. I think it's a good paper that introduces effective and efficient video architecture, which will be valuable to the video research community. Therefore, I'm sticking with my original recommendation to accept the paper.

---

### Official Review · Reviewer_4ga6 · 2021-10-31

**Correctness:** 3
**Technical Novelty And Significance:** 3
**Empirical Novelty And Significance:** 3
**Recommendation:** 6
**Confidence:** 3

**Main Review:**

Strength:
- The empirical study is sufficient, both the state-of-the-art comparison and ablation study.
- The general idea of taking the best from 3D CNN and vision transformer is interesting.

Weakness:
- The related work is pretty brief, which does well explain the relation w.r.t. to the literature in similar direction. E.g. (Liu et al., 2021b).
- The phrase "local redundancy" appears quite vague to me. According to the paper, the redundancy seems to refer to the pixels within a local patch, which can be reduced by CNN with 3D convolution. For me, what carried by a local patch is more of "local structure" than "redundancy", which should be capture instead of "suppress".
- The paper introduces "DPE" module and the description can barely be found. It is not clear why DPE is able to "dynamically integrate 3D position information".
- Also, the "FFN" module is introduced with only one sentence.
- In shallow layers, the affinity map A_n becomes a learnable parameter matrix a_n. In this case, A_n is fixed for any test video. This sound counterintuitive. Please correct me if I mistook it.


**Summary Of The Paper:**

In this paper, a new architecture, Uniformer, is proposed to learn spatial-temporal pattern in videos.
The new architecture is claimed to effectively aggregate both local information and global information.
In contrast, the previous two main approaches, 3D CNN and vision transformer, can only take care of one aspect.
The main building block for Uniformer is a Multi-Head Relation Aggregator, which behaves differently at shallow layers and deep layers, to aggregate local and global information respectively.
Extensive empirical studies demonstrate the strength of the proposed method.

**Summary Of The Review:**

This paper explores an interesting direction on how to enable transformer-like architecture aware of both local and global information.
The experiments are carefully designed and results are encouraging. The reviewer would appreciate it if the authors address the above concerns.

---

> ### Author Response · Authors · 2021-11-21
> **Response to Reviewer 4ga6**
>
> Thanks for your constructive comments.
> We provide our feedbacks and modify our paper as follows.
> ------
> **Q1: The related work is pretty brief, which does well explain the relation w.r.t. to the literature in similar direction. E.g. (Liu et al., 2021b).**
>
> **A1:**
> Thanks for your suggestion.
> We would like to further clarify the relationship between our UniFormer with Video Swin Transformer [1].
> First,
> both models share a similar insight of learning video representation in a hierarchical manner.
> However,
> different from shifted-window self-attention in Video Swin Transformer,
> we introduce a concise relation aggregator in our UniFormer.
> Since our relation aggregator can be efficiently instantiated by depthwise spatiotemporal convolution in the shallow layers,
> UniFormer achieves a better accuracy and efficiency balance.
> For example,
> UniFormer only has 15\% GFLOPs of Video-Swin-T while achieving 2\% top-1 accuracy improvement on Kinetics400 (Table2, Page 6).
> We have added more discussions with other relevant works in Section 2 (Page 3) and Section 3.4 (Page 6) of our revision.
> ------
> **Q2: The phrase "local redundancy" appears vague to me.**
>
> **A2:**
> Thanks for this comment.
> We use this phrase to emphasize that,
> vision transformers contain redundant computation when capturing local features.
> As shown in Figure 1 (Page 2),
> though TimeSformer compares similarities among all the tokens,
> it finally learns local representations.
> Such redundant self-attention brings large computation costs in the shallow layers.
> To tackle this "local redundancy",
> we introduce our relation aggregator with local token affinity.
> We have clarified this in our revision (Table 1, Page 5).
> ------
> **Q3: The paper introduces "DPE" module and the description can barely be found. It is not clear why DPE is able to "dynamically integrate 3D position information".**
>
> **A3:** Thanks for your constructive comment.
> We have added another subsection (Section 3.3 of our revision, Page 5) to further explain our DPE design.
> Since absolute position embedding is hard to handle different input sizes,
> we extend condition position embedding (CPE) [2] to design our DPE.
> It is instantiated as 3D depth-wise convolution with zero paddings,
> thus it can overcome permutation-invariance and is friendly to arbitrary input lengths.
> Moreover,
> it has been proven in CPE [2] that zero paddings help the tokens on the borders be aware of their absolute positions,
> thus all tokens can dynamically integrate the 3D position information via querying their neighbor progressively.
> ------
> **Q4: The "FFN" module is introduced with only one sentence.**
>
> **A4:**
> Thanks for this comment.
> Since FFN is a standard submodule in vision transformers,
> we do not use much space to describe it.
> As shown in Figure 3 (Page 4),
> FFN consists of two linear layers with one GELU function.
> The channel number will be first expanded by ratio 4 and then reduced,
> thus all token representations will be enhanced.
> We add these descriptions in Appendix B of our revision (Page 14).
> ------
> **Q5: In shallow layers, the affinity map $\mathbf{A}_n$ becomes a learnable parameter matrix. In this case, $\mathbf{A}_n$ is fixed for any test video. This sound counterintuitive.**
>
> **A5:**
> We would like to clarify the misunderstanding.
> As mentioned in Section 3.2,
> video contents between adjacent tokens vary subtly in the shallow layers,
> since the receptive field of tokens is small.
> In this case,
> it is not necessary to make token affinity dynamical in these layers.
> Hence,
> we use a local learnable parameter matrix to describe token affinity,
> which is just relevant to the relative 3D position between tokens.
> Note that,
> it does not mean that our Uniformer is fixed for any test video.
> In the deep layers,
> token affinity depends on dynamical comparison among token features,
> since video contents between adjacent tokens tend to change with larger receptive fields.
> Therefore,
> our UniFormer is still video-dependent.
> ------
> **Reference**
>
> [1] Liu Z, Ning J, Cao Y, et al. Video swin transformer[J]. arXiv preprint arXiv:2106.13230, 2021.
>
> [2] Chu X, Tian Z, Zhang B, et al. Conditional positional encodings for vision transformers[J]. arXiv preprint arXiv:2102.10882, 2021.

---

### Official Review · Reviewer_axKi · 2021-11-04

**Correctness:** 4
**Technical Novelty And Significance:** 4
**Empirical Novelty And Significance:** 4
**Recommendation:** 8
**Confidence:** 3

**Main Review:**

+ Well-written
+ Thorough ablation study, including clear visualization of the attention
+ Clean formulation for the local and global affinity token and how they imitate 3Dconv and the standard transformer model.
+SOTA accuracy with 10x more efficiency compared to strong baselines.

-It would be nice to understand how the method apply to fine grained tasks such as human keypoints or segmentation as transformers are usually not very good in such task (vs. convolution).



**Summary Of The Paper:**

This paper proposes a new transformer model for video understanding task. The proposal algorithm has the benefit of both 3Dconv to efficient capture local context and transformer for global reasoning. The other innovation is the dynamic position encoding. The proposed method is validated on Kinetics and Something something where it set new accuracy record.

**Summary Of The Review:**

Solid paper with clean formulation, experimental analysis, and well written.

---

> ### Author Response · Authors · 2021-11-21
> **Response to Reviewer axKi**
>
> Thanks for your constructive comments.
> We provide our feedbacks as follows.
> ------
> **Q1: How the method apply to fine grained tasks such as human keypoints or segmentation as transformers are usually not very good in such task (vs. convolution).**
>
> **A1:**
> Following your suggestion,
> we adapt our UniFormer for semantic segmentation (ADE20K [1]).
> We simply equip our 2D UniFormer-S with Semantic FPN [2] and follow the same training strategy in PVT [3].
> As shown in the following table,
> our UniFormer-S shows its great power on such dense tasks,
> e.g.,
> UniFormer-S only has 50% parameters of Swin-S [4] while achieving 1.4% mIOU improvement.
>
> |Method | #Param.(M)  | mIoU on ADE20K |
> |:-------------|:-------: |:-----------: |
> |ResNet101 [5] | 47.5 | 38.8 |
> |PVT-L [3] |  65.1 | 42.1 |
> |Swin-S [4] |  53.2 | 45.2 |
> |Our UniFormer-S |  **27.0** | **46.6** |
>
> ------
> **Reference**
>
> [1] Zhou B, Zhao H, Puig X, et al. Scene parsing through ade20k dataset[C]//Proceedings of the IEEE conference on computer vision and pattern recognition. 2017: 633-641.
>
> [2] Kirillov A, Girshick R, He K, et al. Panoptic feature pyramid networks[C]//Proceedings of the IEEE/CVF Conference on Computer Vision and Pattern Recognition. 2019: 6399-6408.
>
> [3] Wang W, Xie E, Li X, et al. Pyramid vision transformer: A versatile backbone for dense prediction without convolutions[J]. arXiv preprint arXiv:2102.12122, 2021.
>
> [4] Liu Z, Lin Y, Cao Y, et al. Swin transformer: Hierarchical vision transformer using shifted windows[J]. arXiv preprint arXiv:2103.14030, 2021.
>
> [5] He K, Zhang X, Ren S, et al. Deep residual learning for image recognition[C]//Proceedings of the IEEE conference on computer vision and pattern recognition. 2016: 770-778.

---

> > ### Comment · Reviewer_axKi · 2021-12-01
> > **Post rebuttal comment**
> >
> > The rebuttal addresses my concerns. I follow my original score assessment and recommend this work for acceptance.

---

### Official Review · Reviewer_Na5h · 2021-11-07

**Correctness:** 4
**Technical Novelty And Significance:** 3
**Empirical Novelty And Significance:** 3
**Recommendation:** 8
**Confidence:** 4

**Details Of Ethics Concerns:**

N/A.

**Main Review:**

Strength:
- The paper makes an insightful connection between X3d's conv block and transformer block, and build a unified computational block. Compared to current methods focus on adding good properties of conv blocks to new transformer blocks (e.g., swin-transformer), the proposed method is more elegant conceptually.
- On the standard video action recognition benchmark, the proposed method achieves SOTA result, which is impressive.
- The paper provides comprehensive ablation studies on the design choices.


Weakness
- Lack of reference of previous works on the relationship between conv layer and transformer block.
- Table 2 and Table 3. The bold highlights are inconsistent. Table 3 highlights the best accuracy results among all methods, while Table 2 highlights results are not so. In addition, highlights in Table 2 may make readers ignore the fact that MoViNet-A6 results are comparable in both efficiency and accuracy.
- Lack of explanation of DPE. Currently, DPE is briefly introduced as an extension of CPE in the subsection of MHRA. However, DPE is not part of MHRA and it'll be better to have it as its own subsection.
- Lack of comparison with MoViNet-A6+transformer. Despite the bigger design space of L and G, the ablation studies show that the CNN+Transformer architecture works the best. Thus, it is unclear if the proposed method can outperform MoViNet-A5+Transformer blocks with similar GLFOPs. This is only a minor comment, as the proposed method already achieves significant enough performance for publication.

**Summary Of The Paper:**

This paper proposes a new computation block that unifies 3D convolution and transformer block for video action recognition. Starting from the transformer block, the core of the proposed block is its attention module which is equivalent to 3D CNN or transformer block depending on the design of the learnable parameters. On the standard action recognition benchmark, the proposed method outperforms prior art in both accuracy and computation efficiency. The comprehensive ablation studies provide insights on the design choices.

**Summary Of The Review:**

This paper proposes a novel computational module that is elegant conceptually and effective empirically. Conceptually, the proposed block bridges the design choices between the 3D CNN and transformer block, which can be influential to future works on designing better structures in the attention module. Empirically, this paper achieves SOTA results in both accuracy and efficiency. Thus, I r

---

> ### Author Response · Authors · 2021-11-21
> **Response to Reviewer Na5h**
>
> Thanks for your constructive comments.
> We provide our feedbacks and modify our paper as follows.
> ------
> **Q1: Lack of reference of previous works on the relationship between conv layer and transformer block.**
>
> **A1:**
> Thanks for your comment.
> We add more references to previous works in Section 3.4 of our revision (Page 6).
> For example,
> the prior works [1,2] try to replace convolution with self-attention,
> instead of integrating them.
> Recent works [3-5] have introduced convolution to vision transformers but ignored spatiotemporal consideration for video understanding.
> In contrast,
> our UniFormer tackles both video redundancy and dependency with an insightful unified framework,
> achieving a preferable computation-accuracy balance for video classification.
> ------
> **Q2: The bold highlights are inconsistent in Table 2 and Table 3.**
>
> **A2:**
> Thanks for your suggestion.
> We correct them and only highlight the best results in our revision.
> ------
> **Q3: Lack of explanation of DPE.**
>
> **A3:**
> Thanks for your constructive comment.
> We have added another subsection (Section 3.3 of our revision, Page 5) to further explain our DPE design.
> Since absolute position embedding is hard to handle different input sizes,
> we extend condition position embedding (CPE) [6] to design our DPE.
> It is instantiated as 3D depth-wise convolution with zero paddings,
> thus it can overcome permutation-invariance and is friendly to arbitrary input lengths.
> Moreover,
> it has been demonstrated in CPE that zero paddings can help to encode token position information progressively,
> which is necessary for video understanding.
> ------
> **Q4: A minor comment about comparison with MoViNet+transformer.**
>
> **A4:**
> Thanks for this comment.
> We have tried to combine Transformer with MoViNet-Like model.
> The model does not converge in the video domain, due to the optimization difficulty.
> We would like to further investigate it in future work.
> Here, we would like to discuss why our UniFormer is preferable.
>
> - As shown in Table 2 (Page 6),
> our UniFormer-S only uses 60\% GFLOPS of MoViNet-A5 while achieving competitive accuracy.
> It clearly shows a better accuracy and efficiency balance in our Uniformer.
>
> - MoViNet requires large-scale model searching with expensive computation resources,
> while UniFormer can be trained in the lab without any difficulties.
>
> Hence,
> we believe Uniformer would be a more practical choice for video understanding community and beyond.
>
>
> ------
> **Reference**
>
> [1] Cordonnier J B, Loukas A, Jaggi M. On the relationship between self-attention and convolutional layers[J]. arXiv preprint arXiv:1911.03584, 2019.
>
> [2] Ramachandran P, Parmar N, Vaswani A, et al. Stand-alone self-attention in vision models[J]. arXiv preprint arXiv:1906.05909, 2019.
>
> [3] Wu H, Xiao B, Codella N, et al. Cvt: Introducing convolutions to vision transformers[J]. arXiv preprint arXiv:2103.15808, 2021.
>
> [4] Dai Z, Liu H, Le Q V, et al. CoAtNet: Marrying Convolution and Attention for All Data Sizes[J]. arXiv preprint arXiv:2106.04803, 2021.
>
> [5] Srinivas A, Lin T Y, Parmar N, et al. Bottleneck transformers for visual recognition[C]//Proceedings of the IEEE/CVF Conference on Computer Vision and Pattern Recognition. 2021: 16519-16529.
>
> [6] Chu X, Tian Z, Zhang B, et al. Conditional positional encodings for vision transformers[J]. arXiv preprint arXiv:2102.10882, 2021.

---

### Decision · Program_Chairs · 2022-01-20

**Decision:**

Accept (Poster)

**Comment:**

The paper presents an approach for spatio-temporal representation learning using Transformers. It introduces a particular architecture design, which shows an impressive computational efficiency. The reviewers agree that the experimental results are strong, and unanimously recommend the paper for acceptance. The reviewers find their concerns regarding the details of the approach/setting address after the authors' response.

We recommend accepting the paper.